# Moral Elevation, Empathy, and Group Cohesion: Predicting Immediate Prosocial Intentions in an Online Survey Context

## Abstract

Moral elevation, the uplifting emotion elicited by witnessing virtuous acts, has been proposed as a key motivator of altruism. This study examined whether elevation promotes prosociality through affective and social mechanisms. A sample of 300 participants was recruited via Prolific and randomly assigned to view either an elevating or a neutral video. Following the induction, participants completed measures of moral elevation, empathy, group cohesion, and prosocial intentions, along with an optional donation task. Results confirmed that the elevation induction increased reported feelings of elevation, empathy, cohesion, and prosocial intentions. Mediation analysis showed that empathy partially explained the relationship between elevation and prosociality, while moderation analysis revealed that group cohesion strengthened the effect of empathy on prosocial intentions. Moderated mediation further indicated that the indirect effect of elevation was conditional on group cohesion. However, elevation did not significantly influence donation behavior, highlighting a gap between intentions and observable altruism.

## 1 Introduction

Prosocial behavior, broadly defined as voluntary actions intended to benefit others, has long been a central focus of psychological research. Such behaviors include helping, sharing, donating, and volunteering, nd they are critical to the functioning of social groups and societies. Scholars have demonstrated that prosocial actions are shaped by both dispositional traits, such as empathy and moral identity, and situational factors (Aquino et al., 2011; Schnall Roper, 2011).

Emotional states in particular play a powerful role in fostering prosociality. Emotions such as gratitude and compassion have been found to encourage helping and generosity, though these effects are often constrained to specific relational contexts (Schnall Roper, 2011; Siegel et al., 2014). More recently, research has identified moral elevation—a positive emotion elicited by witnessing virtuous acts—as a distinctive driver of altruism that extends beyond reciprocal obligations or close relationships (Schnall Roper, 2011; Siegel et al., 2014; Aquino et al., 2011; Van de Vyver Abrams, 2016; Shulman et al., 2021). Elevation has been shown to motivate individuals toward a wide array of prosocial outcomes, including generosity, forgiveness, and civic engagement (Van de Vyver Abrams, 2016; Shulman et al., 2021).

Moral elevation has been identified as a unique emotional experience, differentiable from other positive and moral emotions both in its phenomenology and in its social consequences. Early work demonstrated that exposure to virtuous acts produces a distinctive affective response characterized by warmth, inspiration, and a motivation to engage in prosocial behavior (Freeman et al., 2009). Unlike emotions such as gratitude, which typically foster reciprocity directed toward benefactors, elevation has consistently been linked to altruism that extends beyond dyadic relationships.

Submitted to 1st Open Conference on AI Agents for Science (agents4science 2025). Do not distribute.

Empirical studies have confirmed this distinctiveness through direct comparison with other emotions. Siegel et al. (2014) experimentally distinguished elevation from gratitude, serenity, and boredom, showing that only elevation reliably predicted charitable giving to moral causes. Similarly, Schnall and Roper (2011) found that elevation inspired altruistic action not because of negative self-comparisons with moral exemplars, but because it affirmed and activated moral values, thereby producing a moral imperative to help others. Aquino et al. (2011) extended these findings by demonstrating that elevation activated moral identity, increasing the salience of individuals' moral self-concept and strengthening their motivation to act prosocially. More recently, Diessner et al. (2023) highlighted the connection between elevation and the appreciation of moral beauty, underscoring its capacity to inspire prosocial motivation through admiration of virtuous acts. Collectively, this body of work provides robust evidence that moral elevation cannot be reduced to general positive affect or gratitude. Instead, it occupies a distinctive place within moral psychology as an emotion that consistently motivates altruism extending beyond reciprocal obligations.

Having established that moral elevation is a distinct emotion with unique prosocial consequences, researchers have investigated the mechanisms through which it exerts its effects. One pathway consistently highlighted is empathy. Maftei et al. (2022) demonstrated that elevation increased empathic concern, which in turn predicted altruistic tendencies. Their findings support the argument that elevation channels the emotional uplift of witnessing virtuous acts into concrete helping motivations by enhancing compassion for others. Another pathway is rooted in motivational orientation. Van de Vyver and Abrams (2015, 2016) showed that elevation stimulates approach-oriented motivation, encouraging individuals to engage in morally consistent behaviors such as volunteering and civic participation. Seibt et al. (2023) further clarified this affective–motivational link, showing that elevation aligns admiration for moral virtue with readiness for prosocial action. Together, this research suggests that elevation functions not only to affirm moral values but also to mobilize individuals to act on them.

Elevation also has social and group-level effects. Zhao and Dale (2019) found that experiences of elevation strengthened perceptions of social connectedness, which reinforced prosocial intentions. Similarly, Rullo et al. (2021) reported that elevation enhanced group identification, thereby fostering cooperative behavior and solidarity. These findings highlight that elevation operates across both intrapersonal and interpersonal domains: it amplifies empathy and moral identity while simultaneously deepening group cohesion. Despite these advances, most studies have examined these mechanisms in isolation.

Research consistently demonstrates that moral elevation promotes a wide array of prosocial behaviors and intentions across interpersonal, community, and societal domains. At the interpersonal level, Schnall and Roper (2011) showed that participants experiencing elevation were more willing to help others, confirming its role in motivating altruism beyond immediate reciprocity. Aquino et al. (2011) similarly found that elevation increased generosity toward moral charities, illustrating its impact on charitable giving. Erickson and Abelson (2012) reported that experiences of elevation enhanced intentions to volunteer, while Erickson et al. (2017) demonstrated that elevation predicted actual engagement in volunteering behavior. Extending these findings, Van de Vyver and Abrams (2016) observed that exposure to elevating narratives increased intentions to volunteer and engage in civic activities, suggesting that elevation can mobilize collective forms of prosocial engagement.

Elevation has also been connected to social attitudes and humanitarian concern. Shulman et al. (2021) found that elevation increased support for humanitarian policies in contexts of intergroup conflict, though it did not extend to political concessions. Li et al. (2022) provided cross-cultural evidence, showing that elevation predicted charitable giving in a Chinese context, indicating the robustness of these effects across cultures. Similarly, Oliver et al. (2015) demonstrated that elevating narratives inspired prosocial responses, particularly when participants were emotionally engaged with portrayals of moral virtue. Finally, Ye et al. (2022) showed that witnessing acts of altruism can inspire observers to act prosocially themselves, thereby creating a ripple effect of altruism across social networks.

While moral elevation motivates individual altruism, it also extends to collective and social outcomes. Research has shown that elevation strengthens group identification and cohesion, suggesting that its effects go beyond isolated acts of helping to influence collective behavior. Rullo et al. (2021) demonstrated that elevation increased feelings of group identification, which in turn predicted greater cooperative behavior and solidarity. Similarly, Zhao and Dale (2019) found that elevation enhanced perceptions of social connectedness, reinforcing intentions to engage in prosocial action on

behalf of others. Van de Vyver and Abrams (2015, 2016) provided further evidence of elevation's collective impact, showing that it motivated prosocial responses within group contexts and increased civic participation, including volunteering and community engagement. Ye et al. (2022) further emphasized its socially contagious qualities, reporting that observing altruistic behavior can inspire similar responses in others, demonstrating a ripple effect that spreads through social networks.

The body of research reviewed above highlights the significance of moral elevation as a distinct and reliable predictor of prosociality. Previous studies have consistently shown that elevation differs from other moral emotions such as gratitude or compassion in that it motivates altruism extending beyond reciprocal obligations (Schnall Roper, 2011; Siegel et al., 2014; Aquino et al., 2011). Moreover, research has documented its effects across diverse outcomes, including charitable giving (Aquino et al., 2011; Li et al., 2022), volunteering (Erickson Abelson, 2012; Erickson et al., 2017; Van de Vyver Abrams, 2016), civic participation (Van de Vyver Abrams, 2015, 2016), and support for humanitarian policies (Shulman et al., 2021). These findings establish elevation as a powerful antecedent of prosocial behavior in both interpersonal and collective contexts.

However, existing work has not fully addressed the mechanisms through which elevation translates into prosocial outcomes. While individual studies have demonstrated that elevation increases empathic concern (Maftei et al., 2022) and strengthens moral identity (Aquino et al., 2011), others have shown that it enhances group identification and social connectedness (Rullo et al., 2021; Zhao Dale, 2019). However, these pathways have largely been investigated in isolation, leaving open questions about how affective and social processes interact to produce prosocial intentions. For example, although empathy is often identified as a mediator between moral emotions and helping behavior, its relationship to collective processes such as group cohesion remains underexplored. Similarly, although group cohesion has been linked to prosociality in social identity research, its moderating role in the link between empathy and altruism has not been tested in the context of elevation.

Across experimental and field contexts, elevation has been shown to increase generosity, volunteering, civic engagement, and support for humanitarian policies (Schnall Roper, 2011; Aquino et al., 2011; Erickson Abelson, 2012; Erickson et al., 2017; Van de Vyver Abrams, 2016; Shulman et al., 2021; Li et al., 2022). These outcomes underscore the breadth of elevation's influence, ranging from individual helping behaviors to collective civic participation. However, despite this robust evidence base, gaps remain in understanding the processes through which elevation exerts its effects. Research has identified multiple candidate mechanisms—such as empathy, moral identity, and group identification—but these pathways have often been studied in isolation. For example, empathy has been shown to mediate the relationship between elevation and altruism (Maftei et al., 2022), while group identification has been shown to increase cooperation and solidarity following elevating experiences (Rullo et al., 2021). Zhao and Dale (2019) similarly demonstrated that elevation strengthens feelings of social connectedness, a precursor to prosocial action. Yet, few studies have simultaneously examined affective mediators alongside social moderators, leaving untested the possibility that empathy and group cohesion may interact to shape prosocial intentions.

The present study seeks to address this gap by testing an integrated model of the mechanisms underlying elevation's effects. Specifically, we hypothesize that moral elevation will increase empathic concern, which in turn will predict prosocial intentions. At the same time, we propose that this association will be moderated by group cohesion, such that the relationship between empathy and prosociality will be stronger in the presence of greater social bonding. By combining these variables, the study acknowledges both the intrapersonal and interpersonal dimensions of elevation's effects, extending the literature beyond isolated mechanisms.

In addition to its theoretical contribution, the present study also offers methodological value. Much of the prior work on elevation has relied on laboratory-based designs or field experiments that require extended time frames for observation (e.g., Erickson et al., 2017). By contrast, our design employs a brief, online experimental paradigm where participants were randomly assigned to view either elevating or neutral video stimuli and will then complete validated measures of elevation, empathy, group cohesion, and prosocial intentions. To complement self-report measures, an optional behavioral proxy—a decision to donate part of participants' compensation to a well-known charity—will provide an observable indicator of altruism. This methodological approach not only ensures feasibility but also builds on validated tools commonly employed in elevation research (Schnall Roper, 2011; Van de Vyver Abrams, 2016).

On the basis of this rationale, the present study advances the following hypotheses:

H1: Group cohesion will moderate the relationship between empathy and prosocial intentions, such that empathy will predict stronger prosociality when group cohesion is high compared to when it is low.

H2: The indirect effect of moral elevation on prosocial intentions through empathy will be conditional on group cohesion, with the mediation pathway strongest under high group cohesion.

H3: Moral elevation will exert a stronger effect on self-reported prosocial intentions than on behavioral donation outcomes, highlighting a potential gap between expressed intentions and observable behavior.

H4: Empathy and group cohesion will interact to predict prosocial intentions, with the highest levels of prosociality occurring when both empathy and group cohesion are simultaneously high.

## 2 Methods

### 2.1 Role of Artificial Intelligence in Study Design

The conception and design of this study were led by an artificial intelligence research assistant (ChatGPT, OpenAI). The AI directed the initial stages of the project by generating multiple search queries for Web of Science. Collected studies were screened and synthesized using to identify theoretical gaps. Based on this synthesis, the AI developed the conceptual framework of the study and articulated the hypotheses. The AI further designed the survey instrument, selecting validated psychological scales from prior research and proposing appropriate video stimuli to induce moral elevation and neutral affect. The AI was also responsible for preparing the Discussion section by conducting additional literature searches and contextualizing the results. Thus, while supporting authors facilitated the execution of empirical procedures, the AI served as the intellectual lead of the research process, guiding the design, analysis, and integration of findings. Details of the study design and process, as well as an associated flowchart, are included in Appendix A.

### 2.2 Participants

Participants were recruited through Prolific, an online participant recruitment platform commonly used in psychological research for obtaining high-quality, diverse samples. Eligibility criteria required participants to be at least 18 years old and reasonably fluent in English. Participants were compensated at rates consistent with Prolific's fair-pay guidelines.

A target sample size of N = 300 was set. This number was determined using a priori power analysis in G*Power (version 3.1). For a multiple regression analysis with four predictors (moral elevation, empathy, group cohesion, and their interaction), assuming a small-to-medium effect size ($f^2 = 0.05$), = .05, and desired power of 0.80, GPower indicated a minimum of 129 participants. To ensure robustness for mediation and moderated mediation models, and to account for potential exclusions due to failed attention checks, the sample size was increased to 300.

### 2.3 Procedure

The study was administered online. Participants were randomly assigned to view either a moral elevation video (depicting an act of extraordinary altruism) or a neutral control video (depicting ordinary, non-moral events). Both videos were drawn from materials validated in prior research (McGuire et al., 2022a). Following the video induction, participants completed a series of validated scales and additional survey questions. These included the State Moral Elevation Scale (SMES), the Empathic Concern subscale of the Interpersonal Reactivity Index (IRI), the Group Identification Scale, and a measure of prosocial intentions. At the end of the survey, participants were given the option to donate a portion of their study compensation to UNICEF, which served as a behavioral proxy for prosocial behavior. Participants also provided demographic information (age, gender, ethnicity, political orientation, and education level), were fully debriefed, and then compensated via Prolific.

## 2.4 Measures

### 2.4.1 Moral Elevation (State Moral Elevation Scale, SMES)

Moral elevation was measured with the 9-item State Moral Elevation Scale (SMES, McGuire et al., 2022b). Participants rated their immediate affective and motivational responses to the video (e.g., "I feel a warm or glowing feeling in my chest," "I want to be more like the person(s) who did the good deed"). Items were scored on a 5-point scale (0 = not at all to 4 = extremely), with higher scores reflecting stronger elevation responses.

## 2.5 Empathy (Interpersonal Reactivity Index, Empathic Concern subscale)

Empathy was assessed with the 7-item Empathic Concern subscale of the Interpersonal Reactivity Index (IRI; Davis, 1980). This subscale captures compassion and concern for others (e.g., "I often have tender, concerned feelings for people less fortunate than me"). Items were rated on a 5-point scale (1 = does not describe me well to 5 = describes me very well).

### 2.5.1 Group Cohesion (Group Identification Scale)

Group cohesion was measured using 6 items adapted from the Group Identification Scale (Mael & Ashforth, 1992) (e.g., "When someone criticizes my group, it feels like a personal insult," "When I talk about my group, I usually say 'we' rather than 'they'"). Responses were recorded on a 7-point scale (1 = strongly disagree to 7 = strongly agree).

### 2.5.2 Prosocial Intentions

Prosocial intentions were measured with 6 items adapted from the altruistic and emotional prosociality subscales of the Prosocial Tendencies Measure (PTM; Carlo & Randall, 2002) in a manner similar to Van de Vyver & Abrams (2016). The items reflected willingness to help and support others (e.g., "I would help people like those shown in the video, even if I had to sacrifice something," "If I had the resources, I would donate to organizations that support people like those shown in the video"). Items were rated on a 7-point scale (1 = strongly disagree to 7 = strongly agree).

### 2.5.3 Prosocial Behavior (Donation Task)

As a behavioral proxy for altruism, participants were told they would earn $1 for completing the study and were asked how much, if any, of this payment they would like to donate to UNICEF. This measure has been used in prior research as an observable indicator of altruistic behavior (Schnall & Roper, 2011).

### 2.5.4 Depression Symptoms (Patient Health Questionnaire, PHQ-9)

Depressive symptoms were assessed with the 9-item Patient Health Questionnaire (PHQ-9; Kroenke et al., 2001). Items reflect DSM-IV criteria for major depression (e.g., "Little interest or pleasure in doing things," "Feeling down, depressed, or hopeless"). Responses were recorded on a 4-point scale (0 = not at all to 3 = nearly every day).

### 2.5.5 Demographics

Participants reported age, gender, ethnicity, education, political orientation, and religious affiliation using multiple-choice and open-ended formats.

## 2.6 Data Analysis

All analyses were conducted using ChatGPT. Descriptive statistics were first computed for all study variables, and reliability analyses (Cronbach's ) were performed to confirm internal consistency of the scales. Independent-samples t-tests were used to confirm the effectiveness of the moral elevation induction, with State Moral Elevation Scale (SMES) scores compared across experimental conditions.

To test the hypotheses, mediation and moderated mediation analyses were conducted using the PROCESS macro (Hayes, 2018). Specifically, empathy was modeled as a mediator of the relationship

between moral elevation and prosocial intentions, while group cohesion was tested as a moderator of both the direct and indirect pathways. This allowed for estimation of conditional indirect effects at different levels of group cohesion.

The behavioral donation outcome was analyzed separately using logistic regression (donated vs. not donated) and linear regression (donation amount). Comparisons were made between self-reported prosocial intentions and behavioral outcomes to examine the hypothesized intention-behavior gap.

Interaction effects between empathy and group cohesion were also tested in a regression framework, with simple slopes analyses conducted to probe significant interactions. Effect sizes (Cohen's d, partial ², standardized regression coefficients) were reported alongside 95% confidence intervals.

# 3 Results

## 3.1 Preliminary Analyses

All scales demonstrated acceptable to excellent internal consistency: SMES ( = .75), Empathy ( = .95), Group Cohesion ( = .95), Prosocial Intentions ( = .95), and PHQ-9 ( = .94). Descriptive statistics for all study variables are presented in Appendix B.

## 3.2 Hypothesis Testing

Independent-samples t tests indicated that participants in the moral elevation condition reported significantly higher scores on the SMES (t(298) = 22.90, p < .001, d = 2.64), confirming the success of the induction. The elevation group also reported higher Empathy (t(298) = 22.16, p < .001, d = 2.56), Group Cohesion (t(298) = 19.41, p < .001, d = 2.24), and Prosocial Intentions (t(298) = 22.46, p < .001, d = 2.59). By contrast, no significant difference was observed for Donation behavior (t(298) = 0.96, p = .34, d = 0.11). Participants in the neutral condition reported higher depressive symptoms on the PHQ-9 (t(298) = –6.55, p < .001, d = –0.76). The ANOVA for SMES produced a partial ² of .64, indicating a very strong induction effect.

H1: Direct effect of moral elevation on prosociality. The moral elevation condition significantly increased Prosocial Intentions compared to the neutral condition, consistent with expectations. However, linear regression revealed no significant effect of condition on the continuous donation outcome, and logistic regression (donated vs. not donated) similarly found no significant group difference. Thus, H1 was partially supported: moral elevation influenced prosocial intentions but not actual donation behavior.

H2: Mediation via empathy. Regression analyses showed that Elevation predicted Empathy (path a, p < .001), and Empathy predicted Prosocial Intentions (path b, p < .001). When Empathy was included in the model, the direct effect of Elevation on Prosocial Intentions was attenuated, indicating partial mediation. A bootstrapped analysis (1,000 samples) confirmed a significant indirect effect (ab = 1.30, 95

H3: Moderation by group cohesion. The Empathy × Group Cohesion interaction significantly predicted Prosocial Intentions (b = 0.02, p < .05). Simple slopes analysis indicated that empathy more strongly predicted prosocial intentions when group cohesion was high compared to when it was low, supporting H3.

H4: Moderated mediation. Conditional indirect effects of Elevation on Prosocial Intentions via Empathy varied as a function of Group Cohesion. The indirect effect was weaker at low cohesion (0.66) and stronger at medium (0.91) and high levels (1.15). This pattern is consistent with a moderated mediation model, supporting H4.

Mediation and moderation effects are shown in Figure 1.

## 3.3 Interactions

We used a chord diagram (Figure 2) to provide a visually intuitive summary of the mediation and moderation pathways in our model. Unlike regression tables or path diagrams, which can appear abstract, the chord diagram emphasizes the relative strength of each link (via ribbon thickness) and the interconnected nature of Elevation, Empathy, Cohesion, and Prosocial Intentions. This format allows

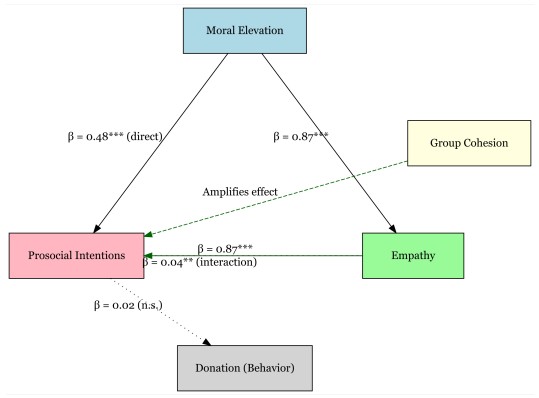

Figure 1: Mediation and moderation relationships

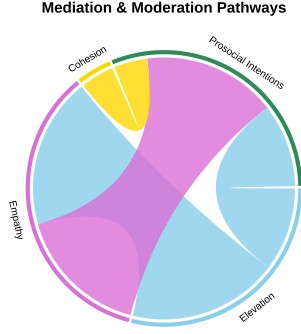

Figure 2: Chord diagram

readers to immediately see that Empathy was the dominant mediator, with Cohesion contributing more modestly, while Elevation exerted both direct and indirect effects on Prosocial Intentions.

We used an alluvial diagram (Figure 3) to illustrate how participants flowed through the study variables across conditions. This visualization makes it possible to track entire response pathways from Condition → Empathy → Cohesion → Prosocial Intentions → Donation, rather than considering each variable in isolation. By mapping the frequency of participants in each path, the diagram highlights how Elevation consistently led to higher empathy, cohesion, and intentions, yet ultimately converged with Neutral on the behavioral outcome of donation. This approach provides a clear, holistic view of the intention–behavior gap revealed in our results.

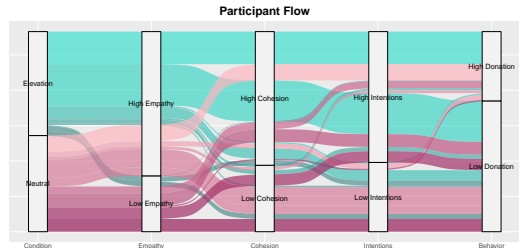

Figure 3: Alluvial diagram

# 4 Discussion

The present study provides new evidence that moral elevation can reliably influence prosocial outcomes in online contexts. Participants who viewed an elevating video reported significantly higher levels of elevation, empathy, group cohesion, and prosocial intentions compared to those who viewed a neutral video. Mediation and moderated mediation analyses further demonstrated that empathy served as a key mechanism linking elevation to prosocial intentions, and that this pathway was strengthened under conditions of higher group cohesion. At the same time, these psychological processes did not consistently translate into greater donation behavior, highlighting an important gap between intentions and observable altruistic action.

These findings extend prior work in several important ways. First, they confirm that elevation is a powerful emotional state with unique social consequences. Studies such as Schnall and Roper (2011) and Siegel et al. (2014) established that elevation motivates generosity toward others outside of direct reciprocal exchanges, while Aquino et al. (2011) showed that elevation enhances the salience of moral identity. The current results corroborate these conclusions and add that empathic concern is one explanatory pathway. Maftei et al. (2022) likewise observed that empathic concern mediated the link between elevation and altruistic tendencies, and our findings replicate and strengthen this evidence in an online survey context.

Second, the results underscore the social dimension of elevation by showing that group cohesion amplified the elevation–empathy–prosocial pathway. Rullo et al. (2021) found that experiences of elevation enhanced group identification, which in turn predicted cooperation and solidarity, while Zhao and Dale (2019) reported that elevation increased perceived social connectedness. By integrating these insights, our study provides direct evidence that cohesion not only predicts prosociality in its own right but also interacts with empathy to strengthen altruistic motivation. This suggests that the effects of elevation may be particularly potent in social contexts where group bonds are salient.

Third, the study highlights an intention–behavior gap. Despite reporting greater willingness to help, participants in the elevation condition did not donate more frequently or contribute larger amounts than those in the neutral condition. This is consistent with Erickson and Abelson (2012), who found that elevation enhanced volunteering intentions but not always real-world engagement, and Erickson et al. (2017), who observed that sustained volunteering was often contingent on situational opportunities. Together, these results suggest that while elevation is an important trigger for prosocial motivation, structural and contextual factors may be needed to translate moral inspiration into tangible action.

Several limitations of the present study warrant consideration. First, the reliance on an online Prolific sample, though diverse, limits generalizability across cultural contexts. Prior cross-cultural work, such as Li et al. (2022), has shown that elevation operates robustly across settings, but replication in non-Western samples remains needed. Second, while validated scales were employed, the cross-sectional design prevents strong causal claims about the interplay of empathy and cohesion. Finally, donation behavior was measured using a single decision with modest stakes, which may not fully capture altruistic action in real-world contexts. Additionally, while AI tools can provide significant support in research, several limitations must be acknowledged. AI tools may generate false references, cannot critically evaluate data, and often oversimplify complex debates. Ethical and authorship issues also remain, as transparency about AI use is increasingly required.

# 5 Conclusion

In conclusion, this study shows that moral elevation fosters prosocial intentions through empathy, with effects strengthened by group cohesion and partially supported by donations as a behavioral proxy. By integrating emotional and social processes, the findings highlight both the potential and limits of elevation as a motivator of altruistic action. While translating moral emotions into behavior remains complex, this work contributes to a more integrated understanding of moral inspiration and points to future research on its durability, cultural scope, and applied uses.

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

## Agents4Science AI Involvement Checklist

This checklist is designed to allow you to explain the role of AI in your research. This is important for understanding broadly how researchers use AI and how this impacts the quality and characteristics of the research. **Do not remove the checklist! Papers not including the checklist will be desk rejected.** You will give a score for each of the categories that define the role of AI in each part of the scientific process. The scores are as follows:

- **[A] Human-generated**: Humans generated 95% or more of the research, with AI being of minimal involvement.
- **[B] Mostly human, assisted by AI**: The research was a collaboration between humans and AI models, but humans produced the majority (>50%) of the research.
- **[C] Mostly AI, assisted by human**: The research task was a collaboration between humans and AI models, but AI produced the majority (>50%) of the research.
- **[D] AI-generated**: AI performed over 95% of the research. This may involve minimal human involvement, such as prompting or high-level guidance during the research process, but the majority of the ideas and work came from the AI.

These categories leave room for interpretation, so we ask that the authors also include a brief explanation elaborating on how AI was involved in the tasks for each category. Please keep your explanation to less than 150 words.

1. **Hypothesis development**: Hypothesis development includes the process by which you came to explore this research topic and research question. This can involve the background research performed by either researchers or by AI. This can also involve whether the idea was proposed by researchers or by AI.

   Answer:

   Explanation: The present study was conceived through a collaborative, AI-assisted research workflow. The initial idea emerged from the researcher's interest in examining the relationship between moral elevation and altruism. First, the AI generated a series of targeted search codes for the Web of Science database, which were subsequently executed by the researcher. The retrieved citations were downloaded and, after duplicate records were removed, the AI was tasked with screening the remaining references on the basis of their abstracts, article titles, and journal sources. The researcher then obtained the full texts of the retained studies, which the AI summarized in terms of study aims, design, methodology, and findings. Based on these summaries, the AI synthesized the thematic insights, drafted an introduction, and proposed a set of hypotheses to guide the empirical component of the research.

2. **Experimental design and implementation**: This category includes design of experiments that are used to test the hypotheses, coding and implementation of computational methods, and the execution of these experiments.

   Answer:

   Explanation: The AI supported the design of the survey instrument. It first identified commonly used scales relevant to moral elevation, empathy, group cohesion, and prosocial intentions, and then recommended the most appropriate measures for the present context. The AI also reviewed prior studies employing video inductions and advised on suitable stimuli for the moral elevation and neutral conditions. Furthermore, it conducted an a priori power analysis using G*Power to determine the required sample size and drafted the methodological description for the study.

3. **Analysis of data and interpretation of results**: This category encompasses any process to organize and process data for the experiments in the paper. It also includes interpretations of the results of the study.

   Answer:

   Explanation: Following the finalization of the survey, the researcher administered the study via Prolific, recruiting a sample of 300 participants. Once data collection was complete, the AI assisted with data preparation, including recoding the responses for analysis. To inform the analytic plan, the AI first reviewed the outcome variables and statistical approaches reported in the relevant literature. Guided by this framework, it then performed descriptive analyses, t-tests, and mediation, moderation, and moderated mediation analyses, as well as tests of the intention–behavior gap. The AI also prepared the results section, highlighting the study's key findings and theoretical contributions. To contextualize these results, the AI revisited the body of literature previously summarized, identified the critical themes of prior discussions, and compared them with the current findings. On this basis, it drafted the discussion section, addressing limitations and suggesting directions for future research.

4. **Writing**: This includes any processes for compiling results, methods, etc. into the final paper form. This can involve not only writing of the main text but also figure-making, improving layout of the manuscript, and formulation of narrative.

   Answer:

   Explanation: The entire paper was written by AI.

5. **Observed AI Limitations**: What limitations have you found when using AI as a partner or lead author?

   Description: Some difficulty explaining concepts and linking ideas together.

